# Activation of the Anaphase Promoting Complex Restores Impaired Mitotic Progression and Chemosensitivity in Multiple Drug-Resistant Human Breast Cancer

**DOI:** 10.3390/cancers16091755

**Published:** 2024-04-30

**Authors:** Mathew Lubachowski, Cordell VanGenderen, Sarah Valentine, Zach Belak, Gerald Floyd Davies, Terra Gayle Arnason, Troy Anthony Alan Harkness

**Affiliations:** 1Department of Biochemistry, Microbiology and Immunology, University of Saskatchewan, Saskatoon, SK S7N 5E2, Canada; lubachow@ualberta.ca (M.L.); zach.belak@proxima-rd.ca (Z.B.); drjerrydavies@gmail.com (G.F.D.); 2Division of Geriatrics, Department of Medicine, University of Alberta, Edmonton, AB T6G 2S2, Canada; 3Department of Anatomy, Physiology and Pharmacology, University of Saskatchewan, Saskatoon, SK S7N 5E2, Canada; cordell.vangenderen@mail.mcgill.ca (C.V.); sav737@mail.usask.ca (S.V.); tgarnaso@ualberta.ca (T.G.A.); 4Division of Endocrinology, Department of Medicine, University of Alberta, Edmonton, AB T6G 2S2, Canada; 5Department of Medicine, University of Saskatchewan, Saskatoon, SK S7N 5E2, Canada; 6320 Heritage Medical Research Centre, University of Alberta, 11207-87 Ave NW, Edmonton, AB T6G 2S2, Canada

**Keywords:** multiple-drug-resistant breast cancer, Anaphase Promoting Complex, cell culture, PDX mouse model

## Abstract

**Simple Summary:**

Patients with treatable cancers all too often return to the clinic with untreatable tumors, requiring highly toxic secondary treatments or palliative care. In this study, we aim to determine whether activation of the Anaphase Promoting Complex (APC) in multiple-drug-resistant (MDR) breast cancer cells will re-sensitize them to chemotherapeutic agents. The APC is required for the targeted degradation of proteins that inhibit passage through mitosis. We found that APC activity is indeed impaired in MDR cells and that chemical activation of the APC increased the sensitivity of these cells to doxorubicin. Mitotic progression was slowed in MDR cells, compared to matched parental drug sensitive cells, with an accumulation of APC substrate proteins. APC activation in nocodazole-arrested cells resulted in increased passage through mitosis with lower APC substrate levels. Mice growing a patient-derived xenografted (PDX) tumor were treated with increasing doses of a chemical APC activator, resulting in a dose-dependent reduction in tumor size. Taken together, our data show that APC activity is reduced in MDR cells, with APC activation resulting in a species- and cancer-type-independent reversal of the MDR phenotype.

**Abstract:**

The development of multiple-drug-resistant (MDR) cancer all too often signals the need for toxic alternative therapy or palliative care. Our recent in vivo and in vitro studies using canine MDR lymphoma cancer cells demonstrate that the Anaphase Promoting Complex (APC) is impaired in MDR cells compared to normal canine control and drug-sensitive cancer cells. Here, we sought to establish whether this phenomena is a generalizable mechanism independent of species, malignancy type, or chemotherapy regime. To test the association of blunted APC activity with MDR cancer behavior, we used matched parental and MDR MCF7 human breast cancer cells, and a patient-derived xenograft (PDX) model of human triple-negative breast cancer. We show that APC activating mechanisms, such as APC subunit 1 (APC1) phosphorylation and CDC27/CDC20 protein associations, are reduced in MCF7 MDR cells when compared to chemo-sensitive matched cell lines. Consistent with impaired APC function in MDR cells, APC substrate proteins failed to be effectively degraded. Similar to our previous observations in canine MDR lymphoma cells, chemical activation of the APC using Mad2 Inhibitor-1 (M2I-1) in MCF7 MDR cells enhanced APC substrate degradation and resensitized MDR cells in vitro to the cytotoxic effects of the alkylating chemotherapeutic agent, doxorubicin (DOX). Using cell cycle arrest/release experiments, we show that mitosis is delayed in MDR cells with elevated substrate levels. When pretreated with M2I-1, MDR cells progress through mitosis at a faster rate that coincides with reduced levels of APC substrates. In our PDX model, mice growing a clinically MDR human triple-negative breast cancer tumor show significantly reduced tumor growth when treated with M2I-1, with evidence of increased DNA damage and apoptosis. Thus, our results strongly support the hypothesis that APC impairment is a driver of aggressive tumor development and that targeting the APC for activation has the potential for meaningful clinical benefits in treating recurrent cases of MDR malignancy.

## 1. Introduction

Cancer is the leading cause of death in Canada, accounting for 28% of all deaths [1,2]. It is estimated that two in five will develop cancer in their lifetimes, with one in four dying from cancer. The 5-year survival rate for all cancers was at 64% from 2015 to 2017, but it varies, with lung cancer survival at 22% and breast cancer survival at 89%. Cancer recurrence is a major issue with almost all cancers. Initial treatment benefits can be met in later life with recurrence of the cancer. The rates differ, with glioblastoma experiencing almost 100% recurrence and estrogen-receptor-positive breast cancers experiencing 5–9% recurrence [3]. Breast cancers (BC) are heterogenous in nature and are typically described as luminal A (estrogen/progesterone receptor (ER/PR)^+^, HER2^−^, Ki-67^−^), luminal B (ER/PR^+^, HER2^−^, Ki-67^+^), luminal HER2 (ER/PR^+^, HER2^+^), HER2 enriched (ER/PR^−^, HER2^+^), Basal-like (ER/PR^−^, HER2^−^, EGFR^+^), and triple-negative (ER/PR^−^, HER^−^, EGFR^−^) [4]. Luminal A has the best prognosis and lowest rate of recurrence, while the triple-negative BC subtype is the most aggressive and most likely to relapse. ER^+^ tumors are the most common, at ~80% [5,6], with the recurrence of these tumors presenting a significant clinical problem globally. After a 5-year survival period, patients with low grade ER^+^ tumors experienced recurrence rates of 10%, whereas those with high grade tumors had a recurrence rate of 17%, after 5–20 years [7]. Recurrent drug-resistant cancers occur for a number of reasons, with a variety of hallmark responses described: decreased expression of drug targets, increased expression of drug pumps and drug detoxification mechanisms, reduced apoptosis capacity, increased DNA repair, and altered proliferation [8]. Although we are gaining advanced knowledge of the variety of resistance mechanisms displayed, we still do not have a clear understanding of how these mechanisms function, why they are induced, nor how to impede them.

An additional mechanism driving MDR that has come to light in recent years is impairment of Anaphase Promoting Complex (APC) activity that is associated with drug-resistant cancer [9,10]. Numerous studies have observed that APC inhibition is linked with aggressive cancer development in vitro and in vivo [11,12,13,14,15]. Decreased APC activity impairs and slows mitotic progression, permitting further mutagenesis through mitotic delays, increasing aneuploidy and subsequent mitotic slippage [16,17,18]. Decreased APC activity via APC subunit mutation, co-activator (FZR1/CDH1) mutations [11,19,20,21], or impaired upstream signaling is associated with genomic instability and MDR onset [22,23,24,25,26]. 

Our recent work supports observations that the APC is inhibited in aggressive cancer cells and that APC activation can reverse drug resistance [9]. We demonstrated that metformin treatment, when combined with cyclophosphamide, doxorubicin (DOX), vincristine, and prednisone (CHOP) chemotherapy, reversed MDR lymphoma in canines in vivo, as it did in vitro [27]; all dogs tested showed reduced expression of markers of MDR and one canine went into remission [9]. We found that tumor samples derived from the canines expressed high levels of all 33 different APC substrate mRNAs that were present on the canine microarray, and that metformin treatment reduced all levels to normal, indicating that metformin induced APC activity. Using OSW canine lymphoma cells selected for MDR, we demonstrated that activation of the APC reduced RNA and protein levels of all APC substrates tested and re-sensitized the canine MDR cells to chemotherapy. Accumulation of APC substrates and presumed APC inactivation was previously described during cancer progression towards more aggressive behaviors and treatment non-responsiveness [20]. Indeed, the accumulation of mitotic specific proteins in G1 is associated with aggressive cancer progression in patient samples [28]. This supports targeting of the APC to increase its activity to manage MDR malignancies potentially through a mechanism enabling prompt (not delayed) entry into mitosis; driving cancer cells rapidly through anaphase while carrying heavy mutational loads results in chromosome instability and appears to be unsustainable, causing mitotic catastrophe and cell death [29,30,31].

The study described here used human MCF7 breast cancer cells selected for resistance to DOX or tamoxifen (TAM). We observed that APC activity is impaired in both DOX- or TAM-resistant human MCF7 cells and that MDR cells required more time to progress through mitosis. Furthermore, we also found that APC mitotic substrates take longer to degrade in mitosis in MDR cells, and begin to accumulate faster as the cell cycle progressed into G1, indicating a strong uncoupling between cell cycle passage into and out of mitosis, and APC E3 activity in this MDR cell line. We also demonstrate that in vitro activation of the APC in MDR-selected cells (i) enhanced the turnover of APC targets in synchronous and asynchronous cells, (ii) recoupled APC activity with cell cycle progression through mitosis, and (iii) increased DOX cell killing. Our observations are consistent with the critical nature of the APC in protecting cells from aggressive cancer behavior, and confirms that its influence extends beyond a single cancer type, a single species, or a single chemotherapy class.

## 2. Materials and Methods

### 2.1. Cell Lines and Materials

MCF7 human breast cells were obtained from American Type Culture Collection (ATCC) in Manassas, VA, USA. Cells were cultured in 75 cm tissue culture flasks (Corning) in a humidified atmosphere (5% CO_2_) at 37 °C. MCF7 cells were cultured in DMEM high glucose media (Gibco, Grand Island, NY, USA) with 10% FBS and penicillin–streptomycin (Gibco). Doxorubicin hydrochloride (DOX; Pfizer, Manhattan, NY, USA), tamoxifen (TAM; Cayman Chemical, Ann Arbor, MI, USA), APCIN (Sigma), Mad2-Inhibitor 1 (Cayman Chemical), nocodazole (Sigma), and thymidine (Sigma-aldrich cat # t1895) were acquired from the indicated providers. All treatment compounds were reconstituted in dimethylsulfoxide (DMSO). Drug treatments were applied at the concentrations and times as indicated. Flow cytometry was performed as described previously [32].

### 2.2. DOX Selection of MDR Cell Lines

MCF7 parental cells were selected for drug resistance as previously described [33], with initial selection in the presence of 1 µM DOX for 48 h. Following this treatment, the cells were washed with sterile PBS and allowed a 3-day recovery period. Drug resistance selection pressure was then reapplied to the cells by subculturing in the presence of 100 nM DOX for 2 weeks with fresh media changes every 3 days. Following the selection period, drug resistance was verified by MDR-1 Western blot analyses and using a cell proliferation assay that relies on the reduction of the yellow MTT (3-(4,5-dimethylthiazol-2-yl) 2, diphenyl-tetrazolium bromide) to a purple MTT-formazan by mitochondrial reductases, as previously described [33]. Cancer cells were cultured in 6-well multi-well plates in phenol red-free medium to avoid interference with the analysis of the purple formazan product. The formazan product and spectrophotometric analysis was performed at 570 nm. Cells were also assessed for cell viability using the Trypan Blue assay, as previously described [27].

### 2.3. Western Blot Analysis

MCF7 cells were washed once with sterile PBS and harvested using a rubber cell scraper as previously described [34], with the following changes: cells were pelleted via centrifugation at 1000 rpm and resuspended in ice cold RIPA buffer (150 mM NaCl, 50 mM Tris-HCl pH 7.4, 1 mM EGTA, 1% NP-40) with protease and phosphatase inhibitors. The cell suspensions were then sonicated with a 70% duty pulse sonication cycle, and centrifuged to remove cell debris. The antibodies used in this study, typically at a 1:1000/2000 dilution, included APC1^tot^ (Abcam133397), APC1^S355phos^ (Abcam10923), CDC20 (PA5-34775), FZR1/CDH1 (Sigma), CDC27 (Abcam10538), Cyclin B1 (Sigma), HURP (Abcam70744, Proteintech), Securin (Abcam79546), MDR-1 (Sigma), BCRP (Santa Cruz Biotechnology, Dallas, TX, USA; SCBt), TFPI (Abcam, Cambridge, UK), PARP (Sigma), γH2AX (NovusBio, Centennial, CO, USA), histone H3^K9Ac^ (Millipore, Burlington, MA, USA), histone H3^S10phospho^, histone H3^tot^ (Millipore), GAPDH (Millipore), and tubulin (Sigma). Following primary antibody incubation overnight at 4 °C, the blots were probed with a 1:10,000 dilution of a horseradish peroxidase (HRP) secondary antibody. 

### 2.4. Coimmunoprecipitation

For coimmunoprecipitation experiments, media was removed, and the dish was rinsed with 3 mL of PBS. Ice-cold RIPA buffer with 0.1% Triton-X-100 and protease inhibitors was then added to the culture dish and incubated on ice for 20 min. Cells were then scraped into the RIPA buffer using a rubber policeman and underwent the same 70% duty pulse sonication cycle described above. Protein concentrations were determined using a Bradford assay, with final stocks adjusted to 750 μg of protein/mL in RIPA buffer. Cell lysates were first cleared by incubation for 60 min with Protein A sepharose beads. The mixture was then centrifuged and the supernatant was incubated overnight at 4 °C with antibody against CDC27. Next, 15 μL of lysate was collected before primary antibody incubation to use as the “Input” sample. Following overnight incubation with the primary antibody, samples were incubated for 2 h with Protein A sepharose beads. Beads were then centrifuged and the supernatant was collected as the “Unbound” sample while beads were collected as the “Bound” sample. All samples were combined with 2x electrophoresis buffer (4% SDS, 20% glycerol, 10% 2-mercaptoethanol, 0.004% bromophenol blue, 0.125 M Tris HCl, pH 6.8) and boiled before being analyzed using Western blots, as described above.

### 2.5. Animals

As previously described [27], 8 to 14 week-old female NOD/SCID/common gamma-chain knock-out (NSG: NOD/Prkdc^SCID^/IL2RN^−/−^) mice were obtained from Jackson Laboratory (Bar Harbor, ME, USA). All experiments were approved by the University of Saskatchewan animal ethics office, Saskatoon, SK, Canada, in accordance with the guidelines of the Canadian Council on Animal Care. 

### 2.6. Murine Xenograft Experiments

We obtained written informed consent from a patient with TNBC for a tumor sample, as described previously [27], which was compliant with the Research Ethic Board approved protocol at the University of Saskatchewan. We passaged a fragment from the original tumor 8 times in NSG mice prior to use in experiments. The resultant tumor was excised, chopped up into ~2 mm fragments, which were then frozen for future use. Fragments were then grafted subcutaneously into 12 separate NSG mice. Once palpable, mice were injected using intraperitoneal (i.p.) injections with 0, 5, 10, or 25 mg/kg M2I-1. This was defined as day 0. Tumor size was measured every 2 days until day 8 using calipers. Tumor volume was determined using the following equation that considered length, width, and height: V = 4/3 π(L/2)(W/2)(H/2) [35]. Tumor sizes were normalized to the size of the untreated tumor at day 0 of the experiment, which defined when M2I-1 was added. The mice were sacrificed after day 8, with tumors surgically removed and analyzed (n = 3 per treatment arm).

### 2.7. Cell Cycle Arrest

Cell cycle arrests of MCF7 parental and TAM-resistant cells were performed in the presence or absence of M2I-1 pretreatment for 24 h. Cells were cultured to 40% confluence before arresting in 100 nM nocodazole for 16 h. TAM-resistant cells were cultured in the absence of TAM pressure for one week prior to arrest. Cells arrested in the presence of 20 µM M2I-1 were pretreated for 24 h before addition of nocodazole, and M2I-1 was maintained until sampling. Samples were harvested for Western blot analyses and flow cytometry every hour upon washing twice in PBS and releasing into nocodazole-free media. Cells were arrested in S phase using a double thymidine block. Cells were treated twice with 2 mM thymidine for 18 h, with a 9 h break in between, according to published methods [36]. Cells were viewed with an Olympus BX51 fluorescence microscope 100× objective equipped with an Infinity 3-1 UM camera. Images were collected using Infinity Analyse software version 5.0.

### 2.8. Flow Cytometry

MCF7 cells were harvested from 6-well plates via dissociation with 0.25% trypsin-EDTA (Gibco). An equal volume of culture media was added to inactivate the trypsin, and cells were fixed by the addition of 1/10 volume 37% formaldehyde solution (Sigma) with gentle agitation at room temperature for 10 min. Cells were pelleted by centrifugation at 300× *g* for 5 min, and washed by resuspension in distilled water passing through a twenty-gauge needle 5 times to eliminate clumping. The washing process was repeated 3 times. Finally, cells were resuspended in 70% ethanol for storage. Immediately prior to analysis MCF7 cells were stained in the dark for 30 min using Vybrant DyeCycle Violet Stain (Invitrogen, Waltham, MA, USA). Once cells were stained, they were pelleted and resuspended in distilled water before analysis using a Beckman Coulter Cytoflex flow cytometer (University of Saskatchewan, College of Medicine, Core facility, Saskatoon, SK, Canada). Flow cytometry data were analyzed using version 2.5.0.77 of the CytExpert^®^ software. Twenty-five thousand events were recorded for each sample and events with a front scatter versus side scatter fluorescence intensity between 0.8 and 1.2 were selected for further analysis (intact cells have a FS/SS ratio near 1, while debris and cell clumps have higher ratios enabling those events to be rejected). G1 and G2 peaks were selected and the percentage of cells in each peak were determined using the in-built statistics function.

### 2.9. Statistical Analysis

Statistical analyses was performed using a Welch’s paired *t*-test. The non-parametric Kruskal–Wallis Test and the FDR post hoc test were used for multiple comparisons. Statistically significant differences are noted within their respective figure legends. Error bars define the standard error of the mean.

## 3. Results

### 3.1. APC Activity Is Impaired in Drug-Resistant MCF7 Human Breast Cancer Cells

To assess whether the restoration of chemosensitivity in MDR cells following APC activation was independent of species and cancer type, human MCF7 ER^+^ breast cancer cells were selected for resistance to TAM according to our published methods [27,33]. As shown in Figure 1A, cells selected for resistance to TAM alone were more resistant to both 5 mM TAM and 1 mM DOX than the parental cells. To assess APC activity in these matched cell lines, we began by measuring APC1 phosphorylated at serine 355, a key marker of APC activation [37,38]. APC1 must be phosphorylated in order for CDC20 co-activator recruitment to the APC at mitosis. We show, using APC1 serine 355 (APC1^S355ph^)-specific antibodies, that APC1 phosphorylation is reduced in MCF7 cells selected for resistance to TAM (Figure 1B,C). Notably, these MCF7 parental and resistant cells treated with an APC chemical activator (M2I-1; [39]) or an APC chemical inhibitor (APCIN; [40]) demonstrated that APC1^S355ph^ was indeed downregulated when treated with APCIN and elevated when treated with M2I-1 (Figure 1B,C). In Figure 1C, the phosphorylated versus total APC1 (APC1^S355ph^:APC1^tot^) ratio was determined after all bands from three separate experiments were scanned, normalized to the GAPDH load control, and then plotted. It is clear that the inherent level of APC1 phosphorylation in MCF7 chemo-resistant cells is similar to that in unselected parental cells treated with APCIN, and that the level in resistant cells returns to parental levels when treated with M2I-1. This is consistent with our findings that M2I-1 exposure reduced APC substrate protein levels, a marker of APC activation, in canine OSW lymphoma cells selected for resistance to DOX [9]. This also suggests that the APC defect is fully reversible, which is clinically important when considering this as a potential treatment target.

Next, we measured the degree of recruitment of the CDC20 coactivator to the APC in parental versus MDR cells; CDC20 interacts with the APC subunit CDC27 upon APC1 phosphorylation at mitosis and contributes to APC activation [41]. We immunoprecipitated CDC27 from MCF7-sensitive and -resistant cells and measured the relative amount of CDC20 that was coimmunoprecipitated (Figure 1D). As a control we used antibodies against CDC27 to show that similar amounts of CDC27 were immunoprecipitated from both resistant and parental cells. Antibodies against CDC20 demonstrate that markedly less CDC20 was found associated with CDC27 in the resistant cells. We then used antibodies against the second APC co-activator, FZR1/CDH1. Similar to CDC20, there was less FZR1/CDH1 associated with CDC27 in TAM-selected cells. Figure 1E, left panel, shows the quantification of the CDC20-bound lanes in three repeats of the experiment shown in Figure 1D. Figure 1E, right panel, shows the quantitation of the amount of FZR1/CDH1 pulled down with CDC27 antibodies in sensitive and resistant cells. Furthermore, we used antibodies against protein markers of MDR (BCRP, MDR-1, and TFPI) [9,27,33,34] and DNA damage (γH2AX) [42] to confirm that the TAM-selected cells were indeed MDR and experiencing higher levels of DNA damage (Figure 2A–C; two separate experiments performed independently are shown in A and C). Our data shown here indicate that the reduction in CDC20 and FZR1/CDH1 interactions with CDC27 are not cell-cycle-specific and reflect an impairment of both APC^CDC20^ and APC^CDH1^.

We then compared the relative abundance of APC degradation substrates in matched MCF7 parental and chemo-resistant cells. If the APC is specifically impaired in MCF7-resistant cells, then APC substrates should accumulate compared to parental cells. Consistent with this, multiple APC substrate proteins (CDC20, Cyclin B1, HURP, and Securin) were elevated in TAM-selected cells (Figure 2A–C; the replicate lanes in Figure 2A were quantified, normalized to their load controls, and plotted with SEM, as shown in Figure 2B). Next, we assessed MDR and APC substrate protein levels in DOX-selected MCF7 cells to ensure that it was not a drug-specific effect, as TAM and DOX are unrelated first line therapeutics for breast cancer. As anticipated, higher levels of MDR protein markers (TFPI) were observed (Figure 2D; triplicate and duplicate lanes in Figure 2D were scanned, quantified, and plotted, with SEM shown in Figure 2E), as shown previously [27,33,34]. Furthermore, higher APC substrate protein levels (CDC20 and Cyclin B1) were present. Taken together, our observations strongly support the hypothesis that APC activity is impaired in human MCF7 breast cancer cells selected for resistance to unrelated chemotherapeutic agents. This suggests that our observation that APC activity is reduced in canine MDR cancer cells in vitro and in vivo [9] is not canine specific, but potentially a common or recurrent feature of MDR cancer cells. 

### 3.2. APC Activation In Vitro Slows MDR Cancer Cell Growth and Restores APC Substrate Protein Levels to Normal

We have previously shown that APC activation in vitro reduced APC substrate protein levels in chemo-resistant canine lymphoma cells, and re-sensitized them to DOX [9]. Here, we exposed DOX-resistant MCF7 breast cancer cells to increasing doses of the APC activator, M2I-1, and assessed APC substrate levels and changes to relative DOX resistance. Western blot analysis for levels of the APC substrate HURP (Figure 3A) shows that it is elevated in selected cells compared to matched parental cells, and the levels in MDR cells return to parental levels at the highest M2I-1 dose used. We also noted that M2I-1 can reduce protein levels of the APC target cyclin B1 in MCF7 parental cells, but a higher dose of M2I-1 is required (Figure 3B). Next, we measured the viability of MCF7 parental and TAM-selected cells, using MTT, following pretreatment of cells with the doses of M2I-1 shown for 18 h, followed by 48 h of DOX exposure at levels previously determined to differentiate between chemo-sensitive and chemo-resistant cells (Figure 1A). Importantly, M2I-1 alone did not impact cell viability at the concentrations used, but when combined with DOX it restored chemosensitivity, as demonstrated by the reduced viability of MCF7^TAM^ cells, which was comparable to that of parental sensitive cells (Figure 3C). 

### 3.3. In Vivo APC Activation in Tumor-Bearing Mice Stalls Tumor Growth

We have described our patient-derived MDR breast cancer tumor tissue, 4–28, that reliably grows as a xenograft (PDX) in mice (NOD/Prkdc^SCID^/IL2RN^−/−^) [27]. In the experiment shown in Figure 4, 4–28 tumor slices taken from a mouse xenograft tumor were implanted into new mice and monitored until palpable. Each mouse received 1 dose of either mock, 5, 10, or a maximum of 25 mg/kg M2I-1 via intraperitoneal injection, with tumor size measured every 2 days for 8 days (n of 3 per treatment arm). We observed that growth of the 4–28 tumor proceeded unencumbered in mock-treated mice (Figure 4A), whereas there was a dose-dependent decrease in tumor growth with M2I-1 exposure; notably, the highest M2I-1 dose blocked further tumor growth (Figure 4A; quantification and statistical analysis shown in Figure 4B). To confirm that APC activity was increased with M2I-1, we measured APC target degradation in excised tumor and liver tissue from both mock- and 25 mg/mL M2I-1-treated mice. Western blot analyses with antibodies against APC substrates Cyclin B1 and FZR1/CDH1 were performed. Regardless of whether MDR (tumor) or normal tissue (liver) was used, we observed a decrease in Cyclin B1 with M2I-1 treatment. The lower Cyclin B1 band is likely a cleaved Cyclin B1 band as described previously [43]. On the other hand, FZR1/CDH1 decreased only in MDR tissue and not in liver tissue. In conclusion, within MDR tumors, both Cyclin B1 and FZR1/CDH1 substrate levels were decreased with M2I-1 treatment, an indication that the APC was activated.

We also assessed these tumor samples for evidence of apoptosis (via PARP cleavage) [44] and cell killing (via H3^K9Ac^) [45] following APC activation by M2I-1 (Figure 4C). We noted that the apoptosis present in mock-treated tumors was significantly increased upon M2I-1 treatment, in the absence of any chemotherapy, whereas liver tissue had no signal. Similarly, the cell killing biomarker, H3^K9Ac^, appeared only in MDR tumor tissue after M2I-1 exposure (Figure 4C). Together, this analysis implies that the APC was activated by M2I-1 in the tumors grown in mice, leading to increased apoptosis, cell killing, and stalled tumor growth. Importantly, while the systemic dose of M2I-1 used in this analysis was detrimental to tumors, it did not have any obvious impact on the molecular markers measured in the normal liver tissue and after 8 days, and the mice did not display any overt negative signs of the treatment. This is consistent with the nontoxic nature of M2I-1 in vitro in our hands (Figure 3C).

### 3.4. APC Substrate Degradation Is Delayed during Mitosis in Drug-Resistant Cells

To determine if impaired APC function correlates with a delay in mitotic passage in MDR cells, as a means of permitting abnormal and/or damaged cells to repair unsustainable DNA damage [29,46,47], we compared cell cycle progression and cycle positioning over time, as well as APC activity in synchronized cells from both sensitive and resistant MCF7 cell lines. First, we arrested cells in mitosis (100 nM nocodazole) or in S phase (a double thymidine block) to observe levels of APC substrates in sensitive and resistant cells (Figure 5A). We observed that HURP and CDC20 were at their peak levels in mitosis, as expected [20,48], with MCF7^TAM^ mitotic cells exhibiting higher levels than MCF7^Sens^ cells (quantitation of Figure 5A is shown in Appendix A). Next, we arrested parental and TAM-selected MCF7 cells in mitosis (100 nM nocodazole for 16 h), then washed the cells to remove the arresting agent and allow synchronized cell cycle re-entry, with samples taken at the indicated timepoints for up to 24 h for both flow cytometry and APC target quantification, using Western blotting against multiple APC protein substrates. In Figure 5B, we assessed APC substrate level degradation with Western blot analysis from samples taken every 4 h. Overall, substrate levels were higher in MCF7^TAM^ (R) cells when arrested with nocodazole, and HURP levels remained higher for upwards of 8–12 h. At later time points, HURP, CDC20, and Securin all began to accumulate in MCF7^TAM^ cells earlier than in MCF7^Sens^ (S) cells (see Appendix A for quantitation of the Figure 5B), indicating that control of substrate levels entering and exiting mitosis was consistently impaired in selected cells compared to parental MCF7 cell lines. Progression of the cell cycle between parental and selected cells was not obviously different (Figure 5B, lower panel, quantified in Figure 5C), although it appears that TAM-selected cells take longer to exit mitosis and longer to fully enter G1 later in the time course (Figure 5C). Flow cytometry indicates that APC substrates accumulate in MDR cells before a shift from G1 to G2/M, whereas in parental cells, APC substrates have not begun to rise (a complete cell cycle flow cytometry profile over the 24 h time course is presented in Appendix A). Thus, cell cycle progression and APC substrate degradation appear to be temporally uncoupled in selected cells, but are synchronous in parental cells. To confirm that APC substrate degradation is impaired while progressing through mitosis in TAM-selected cells, we assessed multiple samples taken over 8 h from the experiment shown in Figure 5B. This analysis revealed that HURP, CDC20, and Securin levels all remained highly elevated in TAM-selected cells for at least 2 h following release (Figure 5D; see Appendix A for quantification), in contrast to their degradation in parental cells. An extended analysis of this experiment over 18 h assessing CDC20 levels is shown in Appendix A, confirming that CDC20 degradation occurs more rapidly in parental cells than in selected cells following release from a mitotic arrest.

### 3.5. APC Activation Increases Cell Cycle Progression through Mitosis in MCF^TAM^ Cells in Coordination with Enhanced APC Substrate Degradation

To determine if APC activation using M2I-1 can reduce substrate levels during mitosis and realign the cell cycle with APC activity, we pretreated MCF7^Sens^ and MCF7^Tam^ cells with M2I-1 for 24 h prior to nocodazole arrest (M2I-1 treatment continued during the nocodazole arrest), or left cells untreated, then assessed target degradation and cell cycle progression. Following the 24 h M2I-1 pretreatment, samples from unarrested cells (cycling), nocodazole-arrested cells (mitosis), and samples taken hourly for 6 h after synchronized release back into the cell cycle were removed for Western blot and flow cytometry analyses (Figure 6). As expected, untreated asynchronously cycling MCF7^Sens^ and MCF7^TAM^ cells were distributed throughout the cell cycle, and their treatment with nocodazole resulted in a strong M arrest for both (Figure 6A, rows 1 and 3). There were notable differences between MCF7^Sens^ and MCF7^TAM^ cells after M2I-1 pretreatment but before arrest (cycling). It was observed that M2I-1 treatment alone shifted the proportion of MCF7^Sens^ cells towards G1 (Figure 6B, compare cycling cells in panels 1 and 2), suggesting that M2I-1 promotes or accelerates passage through mitosis. M2I-1 pretreatment did not create an obvious change in the cell cycle profile of cycling MCF7^TAM^ cells. The subsequent arrest of these pretreated cells also showed differences, as mitotic arrest of MCF7^Sens^ cells was not as efficient as for MCF7^TAM^ cells, indicating that activation of the APC apparently blunted nocodazole-dependent mitotic arrest in parental MCF7 cells, perhaps by allowing some progression into G1. This inefficiency is not observed in pretreated MCF7^TAM^ cells, where APC is inherently less active.

Synchronized release of untreated cells from mitosis back into the cell cycle also revealed differences between sensitive and MDR cells. MCF7^Sens^ cells began to exit mitosis within an hour of release (Figure 6A, row 1, Figure 6B, panel 1) with continuous cell cycle progression. In contrast, MCF7^TAM^ cells had a prolonged mitotic pause where it took 4 h to begin entrance back into the cell cycle, which then continued with high M content for the remainder of the time course (Figure 6A, row 3, Figure 6B, panel 3). Synchronized release of MCF7^Sens^ M2I-1 pretreated cells from mitosis back into the cell cycle also began within an hour of release, but started with a higher G1 content (Figure 6A row 1 vs. 2), whereas the pretreated MCF7^TAM^ cells showed a slow progression into G1 after 1 h up to the 6 h mark, when it rapidly began to transition into G1 (Figure 6A, row 2 vs. 4, Figure 6B, panel 4), similar to untreated MCF7^Sens^ cells.

Lastly, we determined whether M2I-1 activation of the APC in pretreated cells led to coordinated degradation of APC substrates in alignment with the observed increased rate of passage through mitosis and entrance into G1. Samples were taken from the cells grown for Figure 6A before and at mitotic arrest, and then hourly after synchronized release back into the cell cycle out to 6 h. Western blot analyses of APC substrates were performed to determine differences in target protein abundance between synchronized MCF7^Sens^ and MCF7^TAM^ cells after M2I-1 pretreatment. Three APC targets were assessed over the 6 h experiment: CDC20, HURP, and Securin (Figure 6C). Without pretreatment, MCF7^TAM^ cells harbored higher levels of all target substrates than in MCF7^Sens^ at mitotic arrest (time 0) (Figure 6C–E; as MCF7^Sens^ and MCF7^Res^ were run on separate gels in Figure 6C, see Figure 6D for accumulation of CDC20 in MCF7^Res^ cells). CDC20 degradation relies primarily on APC^CDH1^ [49,50], indicating that APC^CDH1^ is active during this time course since CDC20 levels decline to very low levels by 6 h. The differences in the rate of target degradation were accentuated upon M2I-1 pretreatment in both selected and parental cell lines. Transit through mitosis was faster in pretreated cells when measured by H3Ser10 phosphorylation (Figure 6F), but faster transit through mitosis was evident by measuring area under the curve for G2/M cells only in MCF7^TAM^ cells (Figure 6G). 

Despite the delayed entry into G1 in MCF7^TAM^ cells, we determined that E3-dependent target protein degradation rates in MCF7^TAM^ cells were restored to that of MCF7^Sens^ cells following M2I-1 treatment. The HURP Western blot was repeated and quantified to assess the degradation rate changes following M2I-1 pretreatment in both MCF7^Sens^ and MCF7^TAM^ cells, demonstrating the normalization of target levels in MCF7^TAM^ cells despite initially elevated levels (see Appendix A for quantification of HURP levels in Figure 6C). A different HURP antibody was used in Figure 6C (Proteintech, Rosemont, IL, USA) compared to that used in Figure 5B,D (Sigma, Livonia, MI, USA), revealing high levels of phosphorylated HURP in MCF7^TAM^ cells. HURP phosphorylation is known to require the Aurora kinase, an APC substrate that accumulates in cancer cells [51] and occurs when the APC is inhibited [52]. This adds additional evidence that the APC is inhibited in MCF7^TAM^ cells. We also observed that M2I-1 pretreatment also increased CDC20 and Securin degradation (see Appendix A for quantification of band intensities in Figure 6C). In conclusion, APC activation in MCF7^TAM^ cells normalizes rates of progression through mitosis, increases the degradation of APC substrates to that of parental cells, and enhances cell killing by DOX to match that of parental cells. Given that APC substrate overabundance correlates with more aggressive cancers and less responsive therapy, the reduction in APC targets to normal holds significant potential for therapy.

## 4. Discussion

When multiple-drug-resistant cancer develops, treatment may revert to the use of highly toxic second line chemotherapeutics, or palliative care. There are very few treatment options, if any, that will reverse drug resistance, and certainly no widely used therapy that benefits multiple cancer types. In this study, we show that Anaphase Promoting Complex (APC) activity is low in multiple-drug-resistant (MDR) MCF7 breast cancer cells and that activation of the APC using the small chemical APC activator M2I-1 restores APC activity in resistant cells, recouples cell cycle progression with APC substrate degradation, and re-sensitizes MCF7 cells selected for resistance to tamoxifen (MCF7^TAM^) or DOX (MCF7^DOX^) to levels noted in unmodified parental cell lines. Further, treatment of mice growing PDX triple-negative breast cancer (TNBC) cells with M2I-1 stalled tumor growth, reduced APC substrate levels, and induced PARP cleavage and histone H3 acetylation. Therefore, our prior (canine lymphoma [9]) and current observations that APC activation reverses MDR cancer behavior applies to different cell line models (breast cancer and lymphoma), across species lines (canine and humans), in vitro and in vivo (cell line, canine, and PDX mouse models), and to different methods of selecting cells for resistance (DOX and TAM). We propose that APC function is a general and critical means to maintain cell health and protect against aggressive drug-resistant cancer development. Taken together, our results build a strong case supporting the use of APC activation as a promising means to reverse drug-resistant cancer.

There is ample evidence supporting the idea that normal APC activity protects against cancer development. Many APC subunit mutations have been identified in a variety of spontaneous human cancers [19,21,53,54], which can cause cells to survive exposure to chemotherapy (acquired resistance); mutations in at least seven different APC subunits have been associated with resistance to spindle assembly checkpoint inhibitors [11]. It has been observed that APC impairment is associated with an extended duration of mitosis, allowing time for increased DNA repair, for suppression of chromosome segregation errors, and avoidance of mitotic catastrophe [19], thus, providing a rational mechanism whereby malignant cells survive cytotoxic chemotherapy exposures. In alignment with this idea, we have determined that restoring APC activity in MDR cells results in stalled cancer cell proliferation in vitro and in vivo, and promotes DNA damage and apoptosis (Figure 3 and Figure 4) [9,39,55]. Consistent with the previous literature, we found that passage through mitosis was delayed in MDR cancer cells (Figure 6A,B). It has been suggested that slow-growing cancer cells harboring high loads of chromosome instability use the DNA damage response pathway during mitosis as a genome protective mechanism to survive mitotic catastrophe [29,46,47]. This may, in turn, promote further genomic instability by linking pre-mitotic DNA damage with chromosome instabilities that are then propagated during chromosome segregation. This mechanism may moderate the amount of chromosomal damage carried, as moderate levels of chromosome instability appear to confer treatment resistance and poor prognoses, whereas high or low levels of chromosome instability are associated with better treatment responses [56,57,58]. Therefore, mutations that impair APC function create an environment that is conducive to genomic instability moderation due to slowed mitotic progression.

Another mechanism whereby impaired APC activity may contribute to cancer development, aggressive behavior, and treatment resistance may be due to the failure of pro-oncogenic APC substrates to be appropriately degraded, resulting in a cancer-promoting environment. Multiple APC substrates are known to contribute to cancer development and progression, and have repeatedly been found to be elevated in many cancers, presumably due to reduced APC function and blunting of its E3 activity to target and clear them via ubiquitin-dependent proteasomal degradation (see [20,53] and references therein). APC-targeted proteins, such as CDC20, Securin, HURP, FOXM1, PLK1, and the Aurora kinases, accumulate in multiple unrelated cancer types and are generally associated with more aggressive disease and worse clinical outcomes. In our MDR cell line, we not only confirmed the accumulation of multiple APC substrates (Figure 2), but also noted enhanced HURP phosphorylation in mitosis (Figure 6C), which is attributable to increased Aurora kinase activity [51], although we did not directly demonstrate its protein accumulation.

These protein ‘biomarkers’ of poor prognosis resulted in the development of targeted inhibitors against many APC degradation targets, in isolation, without compelling benefits in patient survival [59]. Aurora kinase inhibitors are currently in phase I–III clinical trials with some success as monotherapy and show promise as a combined therapy, but the inhibitors exhibit high toxicity [60]. Clinical trials using inhibitors against the APC substrate PLK1 have also met with inconsistent results [61,62]. We believe that targeting the root cause, normalizing the APC inhibition present in MDR cells, will be the key to reducing all pro-oncogenic APC substrates and facilitating real clinical benefits to therapy, potentially ones that may be well tolerated. We posit that the APC itself be targeted for activation to normalize the levels of the pro-oncogenic protein degradation targets en masse.

It is important to acknowledge that there is research in the literature demonstrating that APC inhibition, not activation, leads to death of cancer cells in vitro. The APC substrate, CDC20, an APC co-activator in mitosis, is frequently highly overexpressed in different cancer cell lines and human tumors [63,64,65], leading to consideration that elevated CDC20 is an important driver of tumorigenesis, and can serve as a prognostic marker, and a therapeutic target. It is possible that the gene and protein signature of CDC20 elevations and its correlation with more aggressive or metastatic malignancies may be due to CDC20 being the most potent pro-oncogenic APC substrate. This would lead to the possibility that inhibition of just this protein, in a potential background of other elevated substrates, would be sufficient to curtail the growth of these cancer cells. Studies using inhibitors against CDC20 or knockdown of *CDC20* have shown cytotoxicity in vitro [66,67,68]. Similarly, anti-mitotic agents that inhibit APC^CDC20^ result in SAC activation (and, therefore, APC inhibition), delayed or arrested mitosis, and triggered apoptosis in a Bim-dependent manner in vitro [54,69]. Two indirect APC chemical inhibitors work through altering CDC20 binding and activation of the APC: Tosyl-L-Arginine methyl ester (TAME) and APC inhibitor (APCIN). TAME blocks the binding of both APC coactivators, CDC20 and CDH1, to the APC, whereas APCIN binds to CDC20, ultimately impairing the ubiquitination and degradation efficiency of APC substrates (reviewed in [54]). Both inhibitors have anti-tumoral effects [70,71,72,73], despite their different mechanisms of action, and show increased activity when both are used together to create a more potent anti-tumoral effect [40]. 

The observed anti-cancer effect of inhibiting *CDC20* through gene silencing, or chemically through APCIN or TAME, can be interpreted in several ways. First, inhibition of the CDC20 oncoprotein by silencing suggests that, since it is an APC activator, the APC itself must be a critical driver of cancer development. In this case, using an APC activator in cells, such as M2I-1, would be predicted to cause uncontrolled proliferation by pushing compromised cells inappropriately through mitosis. Contrary to this notion, we and others have found that M2I-1 has antiproliferative activity on cancer cells in vitro and in vivo (Figure 3C and Figure 4A) [9,55,74]. Another explanation for why elevated CDC20 levels promote cancer progression is that CDC20 accumulation reflects compromised APC activity, and is not, therefore, able to target CDC20 (or its other targets) for degradation, which is consistent with the overabundance of multiple APC substrates observed in unrelated cancer tissues. While CDC20 may be pro-oncogenic, it is unlikely to act in isolation, as at least 60 of the known 69 human APC substrates are associated with multiple cancer types when they accumulate [20], and are now considered a cancer signature [75,76]. A recent series of papers found that APC substrate mRNAs, including HURP and CDC20, are elevated in multiple cancers, and are now recognized as a hub or signature gene set predictive of poor prognosis cancer (a subset of references are included here; [77,78,79,80]). These substrate accumulations are also associated with more aggressive cancers; in 182 breast tumor samples tested from high grade TNBC, 58% of the samples stained for G1 markers, yet expressed high levels of APC substrates, a cell cycle point when substrates should instead be degraded and at their nadir levels [28]. It has been shown that mitotic slippage can cause this effect where cells bypass a block in mitosis and continue cycling, leading to more aggressive tumors [10]. 

While the optimal use of APC activators and inhibitors in cancer therapy remains unresolved, it is extremely important to consider that the APC is an essential component for normal cell growth, and is necessary for normal cell function. Genetic mouse models lacking either CDC20 or CDH1 are lethal [81,82,83], highlighting the necessity of fine dose management should APC inhibitors, such as APCIN and/or TAME, be considered in the future for human cancer therapy. Conversely, we do not anticipate that APC activation will have the same limitations making dosing theoretically easier; our use of M2I-1 in vitro was not cytotoxic when used alone, yet synergized strongly with DOX to kill MDR cancer cells (Figure 3C). M2I-1 use in vivo also did not obviously impact the health of mice when injected in a short-term experiment (Figure 4). 

## 5. Conclusions

In conclusion, our work supports the hypothesis that APC activation in vitro, in aggressive cancer cells, such as cultured breast cancer cells selected for drug resistance, is sufficient to stall the growth of these cells. We observed that APC activity is reduced in drug-resistant cells (Figure 1 and Figure 2), and that APC activation increases the degradation of APC substrates, re-sensitizing cells to chemotherapy (Figure 3). In our mouse PDX TNBC model, APC activation in vivo, as monotherapy, was sufficient to stall tumor growth (Figure 4), demonstrating that our in vitro results with human (Figure 1, Figure 2 and Figure 3) and canine [9] cancer cells reflect our in vivo situation. A possible underlying mechanism to explain these effects may be through the restoration of mitotic progression and avoidance of mitotic slippage when APC activity is restored. We base this on our observation that, in resistant cells, APC substrates have delayed degradation during mitosis (Figure 5B,D) and that mitotic progression into G1 is slowed (Figure 6A,B). APC activation in MDR cells through pretreatment with M2I-1 normalized progression through mitosis with substrates degraded more rapidly, similar to MCF7^Sens^ cells. Even though the timing of mitotic exit in resistant cells treated with M2I-1 was not fully restored to that of sensitive cells in our hands, it was accompanied with reduced APC substrate levels. This suggests that APC activation recouples mitotic progression with substrate degradation in resistant cells, decreasing the time resistant cells have to manage chromosome instability and survive the next round of division. We suggest that restoration of APC activity may be a general means of killing aggressive cancer cells that is applicable to more than one cancer type, that spans different chemotherapy classes, and may be generalizable given that these observations were consistent across evolutionary boundaries. 

## Figures and Tables

**Figure 1 cancers-16-01755-f001:**
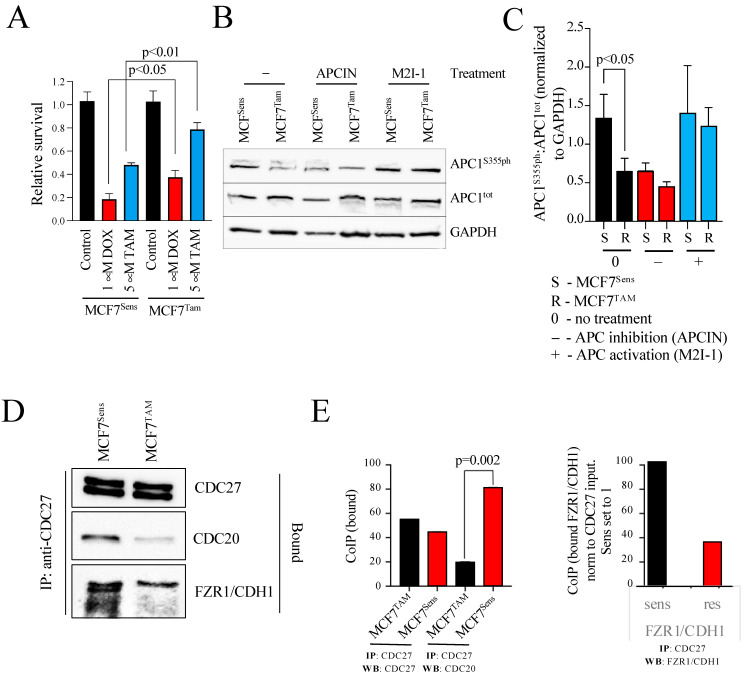
The Anaphase Promoting Complex (APC) is compromised in MCF7^TAM^-resistant cells. (**A**) MCF7 cells selected for resistance to tamoxifen (TAM) are resistant to doxorubicin (DOX), compared to parental MCF7 cells. A two-tailed Welch’s paired *t*-test was used to calculate *p*-values. (**B**) Western blots of MCF7^Sens^ and MCF7^TAM^ cells were performed using 30 µg of protein for each sample and antibodies against either APC1^S355ph^ or APC1^tot^, with antibodies against GAPDH used as a load control. Cells were treated with 5 μM of M2I-1 or 10 μM of APCIN for 18 h. (**C**) The bands from three repeats of the experiment shown in (**B**) were imaged using a VersaDoc. All bands were normalized using GAPDH for each Western blot, with the APC1^ph^/APC1^tot^ ratio determined and plotted. Standard error of the mean is shown. Statistical analyses performed using a paired *t*-test shows a statistically significant decrease (*p* < 0.05) in APC1^S355ph^ in MCF7^TAM^ cells when compared to MCF7^sens^ cells. (**D**) CDC27 was immunoprecipitated (IPed) from MCF7^Sens^ and MCF7^TAM^ cells, with resultant coimmunoprecipitated (CoIPed) proteins (bound) assessed using Western blots (WB) with antibodies against CDC27, CDC20, or FRZ1/CDH1. (**E**) Left panel: protein bands from three CDC27 CoIP experiments followed by westerns with CDC20 and CDC27 antibodies performed in (**D**) were imaged using a VersaDoc,. Bound samples were normalized to input samples and plotted. Standard error of the mean is shown. Statistical analyses performed using a Welch’s paired *t*-test shows a statistically significant decrease (*p* = 0.002) of bound CDC20 in MCF7^TAM^ cells compared to MCF7^Sens^ cells. Right panel: band densities from the CDC27 CoIP followed by westerns with antibodies against FZR1/CDH1 were determined using ImageJ, version 1.53t and plotted. FZR1/CDH1 band intensities were normalized to the CDC27 input. The western was done once. Full uncropped blots are shown in Appendix A.

**Figure 2 cancers-16-01755-f002:**
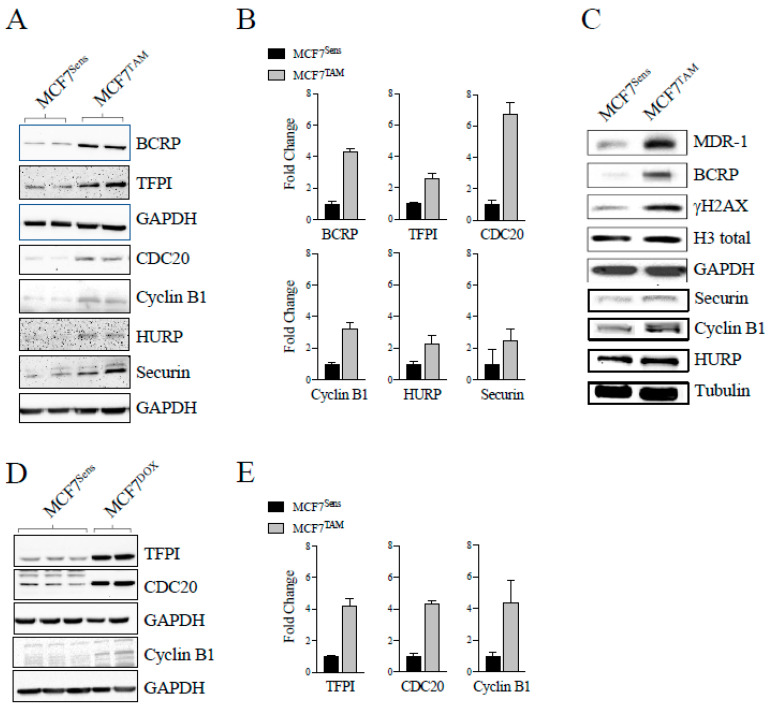
APC substrate proteins are elevated in MCF7 cells selected for resistance to drugs. (**A**) MCF7-sensitive cells and cells selected for resistance to TAM were prepared for Western blot analyses. Antibodies against MDR markers (BCRP and TFPI) or APC substrates (CDC20, Cyclin B1, HURP, and Securin) are shown. (**B**) The duplicate bands shown in (**A**) were scanned, quantified, and normalized to the GAPDH load controls. Average densities of the protein bands in the MCF7^Sens^ cells were set to one. The SEM is shown. (**C**) A second batch of sensitive and MCF7 cells selected for resistance to TAM were prepared for Western blots using antibodies against APC substrates, MDR markers, and indicators of DNA damage. This was performed to demonstrate reproducibility of the method. See Appendix A for quantitation of western blots. (**D**) MCF7-sensitive and cells selected for resistance to DOX were prepared for Western blots using an MDR marker (TFPI) and APC substrate proteins (CDC20 and Cyclin B1). GAPDH and tubulin were used as loading controls. (**E**) The proteins bands in (**D**) were scanned, with the triplicate and duplicate band densities normalized to the load controls, and the average densities determined. The densities for MCF7^Sens^ cells were set to one and fold change shown. The SEM is shown.

**Figure 3 cancers-16-01755-f003:**
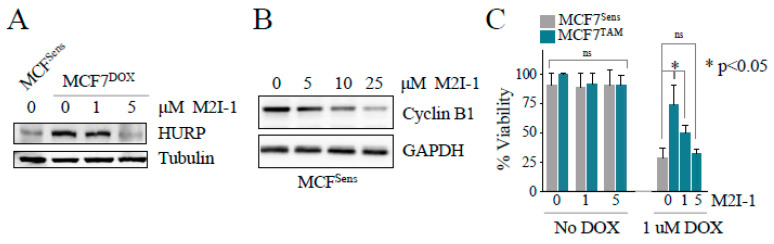
APC activation reduces APC substrate levels and increases DOX-dependent cell killing. (**A**) MCF7^DOX^ cells were treated with 0, 1, or 5 μM M2I-1 for 18 h. Sensitive parental cells were left untreated as a control. Protein lysates from the selected cells and controls were prepared and analyzed using antibodies against the APC substrate HURP. Tubulin was used as a load control. See Appendix A for quantitation of western bands. (**B**) MCF7^Sens^ cells were treated with an increasing dose of M2I-1 as shown for 18 h. Lysates were prepared and analyzed with antibodies against the APC substrate Cyclin B1. GAPDH was used as a loading control. See Appendix A for quantitation of western bands. (**C**) MCF7^Sens^ and MCF7^TAM^ cells were pretreated with 0, 1, or 5 μM M2I-1 for 18 h. MCF7^TAM^ pretreated cells were then exposed to 1 μM DOX for 48 h. Untreated MCF7^Sens^ cells treated with 1 μM DOX for 48 h were used as a control. Cell viability was measured using Trypan Blue. Three biological repeats were performed, with SEM shown. The first 6 columns were tested by one way ANOVA; columns 7 to 10 were tested using a Welch’s two-tailed paired *t*-test. ns; not statistically different.

**Figure 4 cancers-16-01755-f004:**
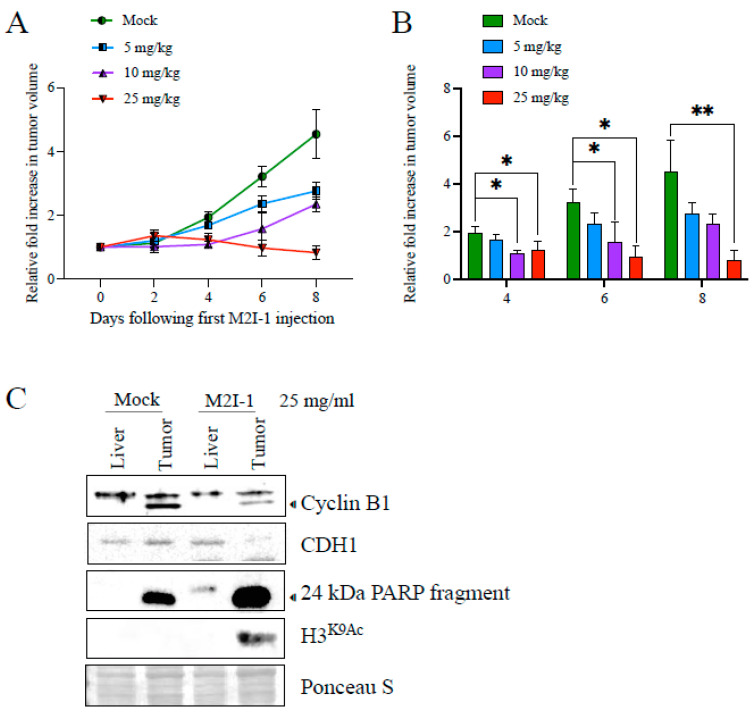
Mice harboring a patient-derived xenografted triple-negative breast cancer (TNBC) tumor treated with M2I-1 show stalled tumor growth with protein alterations indicative of APC activation, induction of apoptosis, and cell killing. (**A**) A patient-derived TNBC tumor sample (4–28 PDX) was engrafted into NOD/SCID mice. Intraperitoneal injections of indicated doses of M2I-1 versus mock (DMSO) were delivered once the tumors were palpable (day 0). Tumor size was measured by caliper every 2 days up to day 8. Tumor sizes were normalized to the untreated tumor at day 0, which was set to 1. n = 3 per treatment arm, SEM shown. (**B**) The data were analyzed using the non-parametric Kruskal–Wallis Test and the FDR post hoc test for multiple comparisons. * *p* < 0.05; ** *p* < 0.005. (**C**) Liver (control) and tumor tissue samples recovered from mock- and M2I-1-treated mice were used for lysate preparation and Western blots using antibodies to assess APC activity (Cyclin B1 and FZR1/CDH1), apoptosis (24 kDa PARP fragment), and cell killing (inhibition of histone deacetyltransferase activity; H3^K9Ac^). Ponceau S was used to show equivalency of loads.

**Figure 5 cancers-16-01755-f005:**
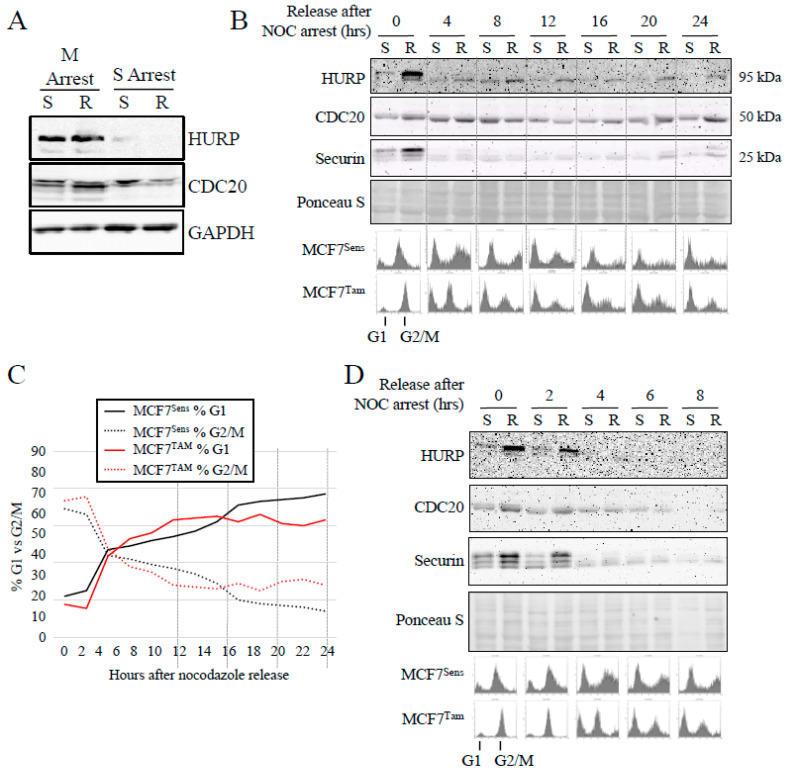
APC substrate degradation is delayed in mitosis in resistant cells. (**A**) MCF7^Sens^ (S) and MCF7^TAM^ (R) cells were arrested with nocodazole (NOC; 100 nM for 16 h) at M or in S phase using a double thymidine block (two treatments of 2 mM thymidine for 18 h, with a 9 h break in between). Protein lysates were prepared and assessed using antibodies against the APC substrates HURP (Sigma) and CDC20. Antibodies against GAPDH were used as a load control. See Appendix A for densitometry of Western bands. (**B**) MCF7^Sens^ (S) and MCF7^TAM^ (R) cells were arrested at M with NOC. The cells were washed and released into fresh media and allowed to re-enter the cell cycle. Samples were removed every 2 h for 24 h for analyses using antibodies against the APC substrates HURP (Sigma), CDC20, and Securin. Ponceau S was used as a load control. Samples were also removed for flow cytometry to determine cell cycle progression following release of cells into fresh media. Selected time points are shown to assess the entire 24 h. See Appendix A for densitometry of Western bands. (**C**) The area under the curve was determined for MCF7^Sens^ and MCF7^TAM^ cells for G1 and G2/M peaks (see Appendix A for complete flow cytometry for the 24 h) following nocodazole arrest and release for 24 h. (**D**) The early samples from the time course presented in (**B**) are shown, as described in (**B**). See Appendix A for densitometry of Western bands.

**Figure 6 cancers-16-01755-f006:**
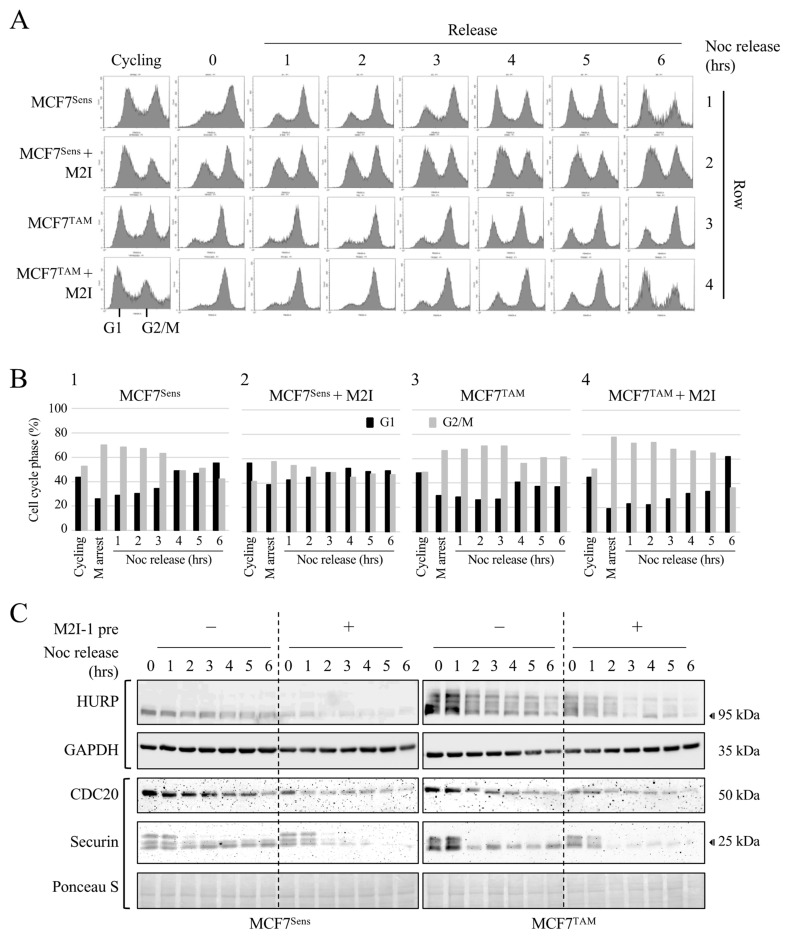
APC activation increases progression through mitosis and APC substrate degradation. (**A**) MCF7^Sens^ (S) and MCF7^TAM^ (R) cells were pretreated with 20 μM M2I-1 for 24 h, or left untreated, before M arrest with 100 nM NOC for 16 h. Cells were washed, then resuspended in fresh media to allow progression through mitosis. M2I-1 was maintained during the release in cells pretreated with M2I-1. Samples were removed every hour for 6 h for flow cytometry. (**B**) The area under the curve was determined for all peaks shown in (**A**) and plotted to quantitatively show cell cycle progression. (**C**) Samples from the cells grown above were removed every hour following release into fresh media for Western blot analyses using antibodies against the APC substrates HURP (Proteintech), CDC20, and Securin. GAPDH was used as a load control for HURP, whereas Ponceau S was used for CDC20 and Securin. One blot was used for the CDC20, Securin, and GAPDH signals since they were of different sizes and easily separatable. (**D**) A single gel was used to compare CDC20, Securin, and H3Ser10^phos^, as a marker of mitosis, in MCF7^Sens^ and MCF7^TAM^ cells. (**E**) All bands for CDC20, Securin, and H3Ser10^phos^ were scanned, quantified, and plotted to compare APC substrate degradation with mitotic progression. (**F**) H3Ser10^phos^ band densitometry was plotted for each cell line, in the presence and absence of M2I-1 pretreatment, to follow mitotic progression following release from NOC arrest. (**G**) The area under the curve for the G2/M peaks was plotted for each cell line, in the presence and absence of M2I-1 pretreatment to follow cell cycle progression through mitosis.

## Data Availability

No new data were created for this study.

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
