# Peer review of "Activation of the Anaphase Promoting Complex Restores Impaired Mitotic Progression and Chemosensitivity in Multiple Drug-Resistant Human Breast Cancer"

_cancers, 2024, doi:10.3390/cancers16091755_

Round 1

Reviewer 1 Report

Comments and Suggestions for Authors

The manuscript was aimed to define whether re-activation of the anaphase promoting complex (APC) can reverse multi-drug resistance (MDR) phenotype in breast cancer cells in vitro and in vivo. 

In general, the manuscript is well preperad and highlights the point about the possibility to enhance sensitivity of chemoresistant cancer cells to certain chemotherapeutic agents (in this particular setting, doxorubicin and tamoxifen) by activating of APC. 

I have the following suggestions and concerns about this manuscript:

1) Since Ki-67 is commonly considered is a proliferative marker, it looks surprising that Luminal A subtype of breast cancer lacks this marker, as was stated in line 50.

2) The authors have to explain how the similar double-time between the  naive and resistant breast cancer cells is possible, because the authors also showed the delay in M-phase for resistant cells when compared with parental cells.

3) The authors are stating about the synergy between APC activation and doxorubicin (e.g., lines 101-102), however the manuscript lacks any data illustrating the synergy score values and appropriate programs (e.g., Synergy Finder, etc.) generally used to deliniate between additive and synergy effects.

4) The authors also often stating about the populations (e.g., lines 112, 113, etc.). Since, the authors did not perform selection and cloning procedures to obtain different subpopulations, it will be more correct to use the term "sublines".

5) Some of the Figures lack statistics to illustrate the differences between the sublines (e.g. Figure 1A).

6) An increased expression of gamma-H2AX in tamoxifen-resistant MCF-7 cells shown in Figure 2B was used as a DNA damage marker. In this particular setting, it is not correct, since tamoxifen does not induce DNA damage. If the authors would like to proof this statement, they have to show similar findings for doxorubicin-reisstant breast cancer cells. Moreover, the extensive DNA damage is a typical feature for sensitive, but not chemoresistant cancer cell lines due to the increased expression of ABC-transporters, activation of DNA repair, etc.

7) Additionally, the increased expression of gamma-H2AX might happen even  without DNA damage and might be due to the mitotic arrest. To deliniate between all these possibilities, the authors have to show DNA damage directly by using Comet assay for naive vs. resistant cancer cells and also illustrate the activation of DNA repair (at least for DSB repair) by providing additional data for the expression of phosphorylated vs total forms of ATM, DNA-PK, ATR, Chk1 and 2 kinases. Otherwise this statement is too preliminary. 

7) Figures 5Band  6A require a quantification for cell cycle distribution.  Moreover, double-staining for M-phase markers (not just of G2/M) is much more suitable for this particular setting. To confirm this general point, the authors have to shown the staining for DNA (PI or 7-ADD) together with phospho-H3 (Ser10) marker.   

Author Response

The manuscript was aimed to define whether re-activation of the anaphase promoting complex (APC) can reverse multi-drug resistance (MDR) phenotype in breast cancer cells in vitro and in vivo. 

In general, the manuscript is well prepared and highlights the point about the possibility to enhance sensitivity of chemoresistant cancer cells to certain chemotherapeutic agents (in this particular setting, doxorubicin and tamoxifen) by activating of APC. 

I have the following suggestions and concerns about this manuscript:

1) Since Ki-67 is commonly considered is a proliferative marker, it looks surprising that Luminal A subtype of breast cancer lacks this marker, as was stated in line 50.

Luminal A breast cancer has low levels of Ki-67. Luminal A subtype cancers grow more slowly, tend to be lower grade, and respond to treatment better (Feeley et al. 2014. Modern Pathology 27:554). No changes to the text have been made.

2) The authors have to explain how the similar double-time between the naive and resistant breast cancer cells is possible, because the authors also showed the delay in M-phase for resistant cells when compared with parental cells.

The reviewer is correct and the language was an error on my part, and was extrapolated from our observation that, at the 24 hour mark, cell cycle progression appeared similar in the experimental strains. We did show a convincing delay in mitotic progression in MDR cells as compared to MCF7, yet because we did not observe a complete cell cycle in the 24 hours of the experiment,  we cannot conclude that the total double times were the same.   We apprecaite that  this is a very crude measure and future experiments will require more robust cell cycling measures. I have removed all reference to similar doubling times in these cells from the text. We have included quantitation of the flow data in Fig. 5C to show that resistant cells do not progress as far into G1 as sensitive cells at the 24 chour mark.

3) The authors are stating about the synergy between APC activation and doxorubicin (e.g., lines 101-102), however the manuscript lacks any data illustrating the synergy score values and appropriate programs (e.g., Synergy Finder, etc.) generally used to delineate between additive and synergy effects.

The reviewer is correct and we appreciate that they highighted our wording. I have rewritten the statement on lines 101-102 to read “…and iii) increased DOX cell killing.”

4) The authors also often stating about the populations (e.g., lines 112, 113, etc.). Since, the authors did not perform selection and cloning procedures to obtain different subpopulations, it will be more correct to use the term "sublines".

We appreciate knowing this terminology, it is important to be precise. I suggest that the best way to refer to these cells is simply call them “cell lines” or “cells”. The TAM and DOX selected cell lines are sublines derived from the parental MCF7 cell line, but calling them “cell lines” or “cells” would be the most consistent naming system in this manuscript.

5) Some of the Figures lack statistics to illustrate the differences between the sublines (e.g. Figure 1A).

We have generated stats for Figure 1A.  We have also included stats for Figs. 3C and 4A.

6) An increased expression of gamma-H2AX in tamoxifen-resistant MCF-7 cells shown in Figure 2B was used as a DNA damage marker. In this particular setting, it is not correct, since tamoxifen does not induce DNA damage. If the authors would like to prove this statement, they have to show similar findings for doxorubicin-resistant breast cancer cells. Moreover, the extensive DNA damage is a typical feature for sensitive, but not chemoresistant cancer cell lines due to the increased expression of ABC-transporters, activation of DNA repair, etc.

Our observation that gH2AX is elevated in TAM resistant cells is consistent with the generation of the resistant phenotype, which is often associated with increased genomic instability and DNA damage. We previously showed that MCF7 multiple drug resistant cells expressed higher levels of gH2AX, compared to drug sensitive MCF7 cells (Davies et al. 2014. PLoS ONE 9:e84611).  We are uusing the marker as an indication of MDR, rather than of TAM exposure.

That said, it has been documented that Tamoxifen can induce DNA damage in both normal and cancer cells, typically inducing DNA strand breaks, including chromosome breaks (Carthew et al. 1995. Cancer Res 55:544; Mizutani et al. 2004. Cancer Res 64:3144; Wozniak et al. 2007. Arch Toxicol 81:519). Detection of gH2AX is now considered a highly specific and sensitive marker for monitoring DNA damage and resolution through repair (Mah et al. 2010. Leukemia 24:679).

7) Additionally, the increased expression of gamma-H2AX might happen even without DNA damage and might be due to the mitotic arrest. To delineate between all these possibilities, the authors have to show DNA damage directly by using Comet assay for naive vs. resistant cancer cells and also illustrate the activation of DNA repair (at least for DSB repair) by providing additional data for the expression of phosphorylated vs total forms of ATM, DNA-PK, ATR, Chk1 and 2 kinases. Otherwise this statement is too preliminary. 

Thank you for bringing this to our attention, our language was not intended to overstep. As above, gH2AX was intended to be used as a marker of the MDR phenotype, given that these MDR cells often carry high loads of DNA damage and genomic instability (Jurkovicova et al. 2022. Int J Mol Sci. 23:14672).  We acknowlege your comment re: mitosis alone is sufficient for elevating this marker, as found in a paper from 2013 (Tu et al. 2013. FEBS Letters 587:3437) that showed that mitotic chromosomes could contain gH2AX in the absence of obvious DNA damage,  suggesting that gH2AX may not always signal DNA damage. Notably, in our TAM resistant cells, the cells are growing asynchronously and are dispersed throughout the cell cycle as shown in Fig. 6A, making  it  unlikely that the observation of increased gH2AX is due to mitotic chromosomes.

Also relevent was the subsequent  paper from Martin et al. (2014. Cell Cycle 13:3026) who tested whether the mitotic gH2Ax signal was indeed independent of DNA damage. They found that Interstitial gH2Ax foci lying on seemingly intact DNA does not seem to mark sites of misrejoining that occurred earlier in the cell cycle and were carried into mitosis. However, they did observe that all gH2AX foci were accompanied by MRE11, a DNA damage response (DDR) factor. They concluded that mitotic gH2AX foci were the result of visible and invisible DNA damage in which at least a partial DDR occurred. It is also true that gH2AX has been associated with events other than DDR, such as roles in brain function (Merighi et al. 2021. Molecules 26:7198).

7) Figures 5B and 6A require a quantification for cell cycle distribution.  Moreover, double-staining for M-phase markers (not just of G2/M) is much more suitable for this particular setting. To confirm this general point, the authors have to show the staining for DNA (PI or 7-ADD) together with phospho-H3 (Ser10) marker.

We have incorporated quantitation for the flow cytometry data. See the new Figs. 5C, 6B, 6F and 6G. We have also included a western showing the levels of the mitotic marker H3Ser10phos as cells progress through the 6 hour arrest release experiment shown in Fig. 6. See new Figs. 6D, 6E.

New text has been added throughout the draft where appropriate to describe these results. A tracked version of the manuscript has been sent to the Editorial board so they can see the extensive editing that has been done to the text and the figures.

Reviewer 2 Report

Comments and Suggestions for Authors

This manuscript shows the relationships of multiple drug resistance (MDR) and Anaphase Promoting Complex (APC) function. The authors previously showed that APC is impaired in MDR cells compared to normal canine control and drug sensitive cancer cells. Therefore, the authors change the cancer model for the confirmation of APC activity with MDR of other cancer in this manuscript. However, the present results are limited. The study needs more examination and revision for the publication.

Comments

Major points

1. In Figure 1D, CDC27 band of IP shows a single band, however, two bands are shown in Figure 1F. Why is the pattern of bands changed? Additionally, E-Cadherin shows smear bands, it is not generally in MCF7 cells. The authors should retake the experiments. And the authors open the raw membrane data of blotting, however, several raw membranes are lacked.

2. In Figure 1E, How is the p-value calculated? The authors showed only two times of experiment. It is not able to calculate in statistics. On the other hand, in Figure 3, 4, and 6, the authors do not perform statistical consideration. The authors should reconsider the statistical discussion in the whole of manuscript.

3. The results and conclusion depend on the single inhibitor, M2I-1, the authors should try another inhibitor in this manuscript. It is difficult to suggest the conclusion by the results which are depend on single molecule, because the off-target effect is not considered.

4. In Figure 4A, the vertical axis was normalized. Therefore, the unit is wrong. It would be better to reconsider the meaning of the units, normalization, and calculation.

6. In Figure 6, the authors should perform the quantification of results.

Comments on the Quality of English Language

Moderate editing of English language required.

There are several typo and mistakes.

Author Response

This manuscript shows the relationships of multiple drug resistance (MDR) and Anaphase Promoting Complex (APC) function. The authors previously showed that APC is impaired in MDR cells compared to normal canine control and drug sensitive cancer cells. Therefore, the authors change the cancer model for the confirmation of APC activity with MDR of other cancer in this manuscript. However, the present results are limited. The study needs more examination and revision for the publication. 

Comments

Major points

  1. In Figure 1D, CDC27 band of IP shows a single band, however, two bands are shown in Figure 1F. Why is the pattern of bands changed? Additionally, E-Cadherin shows smear bands, it is not generally in MCF7 cells. The authors should retake the experiments. And the authors open the raw membrane data of blotting, however, several raw membranes are lacked.

The protein examined in Figure 1D was CDH1/FZR1, an APC co-activator and substrate, not E-cadherin. We have provided all the intact raw blots that we have. Several membranes were cut horizontally so that we could use several antibodies for the same experimental samples. I do not have a full blot for CDH1/FZR1. My interpretation of the smeared banding below full length CDH1/FZr1 may be due to its instability and known degradation. It is unlikely a nonspecific band as it is reduced in the MDR cells similar to the full length CDH1/FZR1 band. The appearance of less of it in MCF7TAM IPs is because there is less pulled down in the co-IP. If it is felt we have not addressed the concerns about CDH1/FZR1 appropriately, and because we cannot provide the intact blot, we can remove the CDH1/FZR1 membrane from Fig. 1D. We apologize for the previous CDC27 blots shown. I have gone over all the co-immunoprecipitation experimental results and have revised the figure. Input samples were not assessed, only bound samples. The experiments and westerns performed with the CD27 and CDC20 antibodies were performed 3 times, while CHD1/FZR1 was only performed once.

  1. In Figure 1E, How is the p-value calculated? The authors showed only two times of experiment. It is not able to calculate in statistics. On the other hand, in Figure 3, 4, and 6, the authors do not perform statistical consideration. The authors should reconsider the statistical discussion in the whole of manuscript.

A Welch’s paired t-test was used to calculate the p-value in Figure 1E. As mentioned above, the Co-ip experiment in 1D was repeated 3 times, with the CDC27 and CDC20 bands quantified and plotted in Fig. 1E. Statistics have been performed in Fig. 3C using one way ANOVAs and Welch’s 2 tailed paired t test. Statistics in Fig. 4B have been performed using the non-parametric Kruskal-Wallis Test and the FDR post-hoc test for multiple comparisons. We did not perform statistical tests for Fig. 6 as the experiment was not repeated 3 times.

  1. The results and conclusion depend on the single inhibitor, M2I-1, the authors should try another inhibitor in this manuscript. It is difficult to suggest the conclusion by the results which are depend on single molecule, because the off-target effect is not considered.

The reviewer is correct to ask for a second APC activator to confirm the results. When we first started this work we used the Spindle Assembly Checkpoint (SAC) inhibitor TTK, which is similar to M2I-1. Inhibiting the SAC results in indirect activation of the APC. Our preliminary results showed that both TTK and M2I-1 had similar results in our experiment. We chose to focus on M2I-1 going forward because M2I-1 has been established as an APC activator in the literature. M2I-1 disrupts the binding of CDC20 to the SAC component MAD2, thereby weakening the SAC and its ability to inhibit the APC and cell cycle progression (Kastl et al. 2015. ACS Chem Biol 10:1661). M2I-1 has also been shown to inhibit cancer growth (Hu et al. 2022. Biomark Res 10:82; Li et al. Cell Div 2019. 14:5). Our results in Fig. 1B also show that M2I-1 increases the APC activation mark, APC1S355phos.  We also discovered a novel peptide that directly binds to the APC and activates it. This peptide results in decreased APC substrate accumulation in MCF7 TAM selected cells and enhances the killing effect of Doxorubicin in vitro. This peptide was discovered using a yeast 2-hybrid screen using Apc10 as bait. In yeast, the peptide activates the APC and increases lifespan. M2I-1 also increases lifespan in yeast. Manuscripts describing this effect are in preparation for both the yeast and the human cell culture work. That is why these additional APC activators have not been used in this paper. We feel that there is enough published and unpublished literature available to conclude that M2I-1 is not functioning through off target effects.

  1. In Figure 4A, the vertical axis was normalized. Therefore, the unit is wrong. It would be better to reconsider the meaning of the units, normalization, and calculation.

It is acceptable practice to normalize data for graphical presentations. What may be at issue is my omission of how we normalized the data and how we calculated tumor volume, which I apologize for. Text in the figure legend and the methods has been added to describe that we normalized all tumor sizes to the size of the untreated tumor at Day 0 and set this to 1. The Y axis represents the normalized tumor volume (mm3). In the methods section we added our tumor size calculation: ? = 4/3 ?(L/2)(W/2)(H/2).

  1. In Figure 6, the authors should perform the quantification of results.

Quantitation of HURP, which was repeated twice is shown in Supplemental Figure 3. Westerns of CDC20 and Securin were only repeated once. These blots have now been scanned and these graphs were added to Supplemental Figure 3. The flow cytometry data has all been quantified by determining area under the peak and plotting. 

Comments on the Quality of English Language:

Moderate editing of English language required.

There are several typo and mistakes.

We observed several typos and mistakes throughout the manuscript. We apologize for not catching all of them prior to submission. If more are noted that we missed, pleased point them out.

Round 2

Reviewer 1 Report

Comments and Suggestions for Authors

The authors responded to my suggestions and comments. The revised  manuscript can be accepted to publication. 

Reviewer 2 Report

Comments and Suggestions for Authors

The authors addressed all my concerns.